# Healthy, Inclusive and Sustainable Catering in Secondary Schools—An Analysis of a Transformation Process with Multiple Tensions

**DOI:** 10.3390/ijerph21030370

**Published:** 2024-03-20

**Authors:** Benjamin Hennchen, Martina Schäfer

**Affiliations:** Center for Technology and Society (ZTG), Technische Universität Berlin, 10553 Berlin, Germany; schaefer@ztg.tu-berlin.de

**Keywords:** school catering, public health, sustainability, inclusion, school food environments, social cohesion

## Abstract

Interest in catering for public sector schools is increasing due to its potential role in addressing the prevailing problems of malnutrition, food insecurity and non-sustainable food habits. Based on the case of secondary schools in Berlin, this study aims to explore this potential by focusing on the process of transformation towards healthy, inclusive and sustainable school catering. It employs a multi-perspective analysis based on the two concepts of food environment and social cohesion. Results are based on quantitative and qualitative data collected via an online survey of pupils from 25 secondary schools in Berlin as well as field notes from six stakeholder events. The survey findings were analyzed by descriptive means and provide explanations for the fact that most of the pupils (66.7%) never eat lunch at school. Based on the qualitative analysis of the stakeholder events, key tensions between actors from the federal state, municipal, school and private levels could be identified. Major areas of conflict arise due to (1) a lack of public funding and catering standards, (2) incompatible demands and preferences, (3) a lack of resources and opportunities for complementary education and participation, and (4) peer and parental influence. Transforming school food environments requires integrative strategies with interventions introduced by multiple actors operating on different levels.

## 1. Introduction

The current state of food production and consumption continues to threaten socio-ecological systems and raises concern about rising malnutrition and obesity rates among adults and children [1,2,3]. The community catering sector is widely regarded by scholars as having the potential to foster necessary large-scale transformation processes towards healthy, inclusive and sustainable food [4,5,6]. In recent years, more attention has been paid to the significant role catering in public sector schools can play in these processes. For instance, studies found evidence that the provision of healthy meals is linked to an improvement in children’s nutritional intake and long-term dietary changes [7,8,9]. There is also evidence of the positive impact of free school meal programs that provide children from different social and economic backgrounds, and especially vulnerable groups, with reliable access to food [10,11,12]. On the supply side, there are analyses of the contribution school catering can make to the development of sustainable value chains by facilitating the consumption of organic and regional food [13,14].

School catering in German public sector schools is criticized for its lack of dietary standards and poor financing [15]. Equally problematic is the high number of pupils who are not willing to eat school lunches on a regular basis [16]. Studies have identified several reasons for the low interest in school meals, such as poor flavors, limited choice of dishes, high prices, and a lack of quality in terms of the food and the physical environment [16,17].

In several other EU countries, school meal programs are in place that aim to improve food access and increase dietary standards as well as the proportion of organic and regional produce [13]. Although studies show no clear results on the effects of these particular programs, there is broad consent among food scholars that providing healthier, sustainable and more inclusive school meals requires less focus on cost efficiency and more on services which promote the well-being of children [4,18]. To reach this objective, particular emphasis is placed on systemic approaches targeting different economic, social, regulative, and cultural factors that influence the daily food choices of children in school. This not only calls for effective food interventions but also for a stronger involvement of all those who play a role in school catering [19]. Although there is growing recognition of these processes and their relevance in the transformation of school food environments, only very few studies so far have presented findings on this topic with particular focus on the German context. Moreover, there has been little investigation of the meal situation in secondary schools, despite the fact that pupils participate even less frequently in school meals in this age group [15].

This study aims to contribute to an understanding of the transformation of school food environments in secondary schools and to identify tensions that might slow down this process. These tensions might arise from the multifaceted process of school meal catering and the complex social relations between the actors involved. These actors include municipal school authorities, caterers, principals, teachers, and pupils, who all have specific roles and responsibilities. This study followed a transdisciplinary research strategy, examining school catering in secondary schools in Berlin. Focusing on Berlin allowed us to cooperate more intensively with practitioners on site and monitor interventions that were initiated during the research. The findings are derived from quantitative and qualitative data and were obtained via an online survey of pupils from 25 secondary schools (amounting to 3015 valid responses) and by analysis of networking events, workshops, and continual consultation with key practitioners. Together, these data provide a comprehensive view of the institutional setting of school catering, including the perspectives of different actors.

## 2. Conceptual Background

The concept of food environments allows us to capture but also differentiate between the manifold aspects underlining the provision of school meals. In general, the concept of food environments describes the setting in which daily food consumption takes place [20,21]. It is usually defined as “the collective physical, economic, policy and sociocultural surroundings” [20] (p. 112). These are the external factors that directly or indirectly determine the food options of people [22]. On the individual level, these factors shape individual consumption patterns by determining what food people have access to, can afford and desire [21].

Based on this concept, a considerable number of studies have presented findings that reveal the impact of school food environments on individual meal consumption. They show that promoting dietary standards and the availability of healthy and sustainable food options is associated with improved eating habits of children [8], a positive change in body weight [23,24] and increased sales of sustainable dishes in schools [25]. Food environments can also have a negative effect on health by providing children with access to unhealthy food options from kiosks or competing fast food restaurants [26].

Furthermore, researchers have identified various factors that discourage or even exclude pupils from eating at school. Examples are limited menu choice, poor food quality, but also short lunch breaks and inconveniences in the lunchroom [15,27,28].

The vast majority of recent studies on the effectiveness of free meal programs confirm that these lead to higher participation rates among pupils and to less social exclusion [11]. Other studies highlight the influence of socio-cultural aspects within and outside schools on meal acceptance. These include the supervisory but also supportive role of teachers [29], the role of eating and relaxing in peer interactions [30], differences in the cultural heritage of pupils [31] and their tastes [32], as well as the influence of meal routines at home [17].

There is a growing interest in approaches that emphasize the importance of engaging school communities in the promotion of healthy [25,33] but also more sustainable food environments [34,35]. The focus is on actors who are directly involved in the operation of school catering. In addition, several authors point out the importance of involving pupils in the process of changing school food environments; for instance, by taking their opinions more seriously and providing them with opportunities to participate in school decision making [36,37,38]. As part of the extended school community, parents have a strong influence on their children’s food consumption at home and in most cases, they also pay for the meals in school [32,39]. Moreover, studies have pointed out that integrating school food into teaching and learning concepts helps children acquire knowledge that can be applied in their daily lives. Aside from traditional school education, school meal services are analyzed in terms of their potential to promote informal learning that provides pupils with positive experiences and stimulates their interest in the topic [40].

Table 1 summarizes the dimensions of school food environments and school communities that have guided the empirical analyses. Firstly, the physical dimension encompasses the availability of food options, kitchen infrastructures and lunchrooms; secondly, the economic dimension encompasses meal prices and the cost of catering and food preparation; thirdly, the regulative dimension encompasses the impact of the legal framework, regulations and dietary standards, as well as the overall organization of food service; fourthly, the socio-cultural dimension encompasses eating habits and pupils’ tastes, as well as the role of teachers and social norms among peer groups.

In order to capture the role of school communities, we refer to the concept of social cohesion. In social sciences, this term has been widely used to describe how individuals are connected to others in society. More generally, it can be characterized by the number and quality of social relations existing within and between individuals, social groups and societies [41,42,43]. In our conceptualization, the analytical focus shifts towards the quality of interactions between the relevant actors on different levels within school catering. Social cohesion also encompasses participation, referring to peoples’ involvement in public discussions or (political) decision making processes that address a common interest [41]. In our analysis, we focus on opportunities for members of the school community to influence school food environments, for instance, by giving feedback or taking part in working groups, programs or committees. Social cohesion also consists of a distributive dimension, which touches on the access to and distribution of resources among community members [41]. The analysis therefore takes into account the inequalities of pupils and parents in terms of their socio-cultural background and available income. At the same time, it captures efforts made to promote inclusiveness in school catering, for instance by providing meals that are both affordable and consider the special dietary needs and restrictions of pupils.

## 3. Materials and Methods

This study followed a transdisciplinary research strategy and was conducted in cooperation with the Networking Center for School Catering (NCSC) (“Vernetzungsstelle Kita- und Schulverpflegung Berlin e.V.”) as well as with the Berlin Senate Department for Education, Youth and Family (SenBJF). The research design makes use of a combination of qualitative and quantitative data.

In consultation with our practice partners, we decided to conduct an online survey of pupils in order to gain insights into the pupils’ perspective on school food environments in Berlin secondary schools. The aim was to address pupils at higher secondary level (8th, 9th and 10th grade), i.e., 14 years old or older.

In the first step, the questionnaire was designed by developing our own research questions but also drawing from categories belonging to a Germany-wide survey on school meals conducted in 2015 [16]. In the second step, the NCSC was involved in the development by giving input on the overall structure and the wording of the questionnaire as well as by proposing their own questions. We mutually agreed with the NCSC on a selection of important questions that were subsequently included in the survey. Three pretests of the questionnaire were conducted to determine whether the pupils understood the questions correctly and that the survey was an appropriate length. In the third step, the pupils’ feedback was incorporated into a final version.

The majority of the questions aimed to analyze different elements of school food environments. These included the pupils’ assessment of the quality and organization of school food services, menu offers and prices, and the perceived quality of the lunchrooms, as well as their general take-up rates, preferences and eating habits (Appendix A). The survey and the questions were ethically approved by the SenBJF, who also contacted the schools.

All Berlin secondary schools were officially invited to take part in the survey by the SenBJF. Of these, 23 public sector and 2 private schools reported back and agreed to take part in the survey. As shown in Appendix A, schools were characterized by different sizes, with numbers of pupils ranging from 150 to 1200. The targeted schools were distributed across eleven of the twelve Berlin districts. In total, 22 of the schools had mandatory afternoon classes, whereas in three cases, attending afternoon classes was optional.

The survey was conducted in cooperation with teachers, who guided the pupils in filling out the questionnaire during or after class and made sure all the questions were understood. Teachers were provided with detailed instructions on the procedure beforehand. Because the survey took place during the COVID-19 pandemic, we needed the assurance that the school canteens were open and operating regularly. Via exchanges with teachers, we obtained additional contextual information on the school food environments, such as meal prices, the canteen situation or the type of food service on offer at the time. All pupils provided informed written consent before they started the survey.

The data were collected from December 2021 to May 2022. Appendix A shows that 3015 valid responses from 1353 female and 1532 male pupils were taken into account in the analysis (108 diverse, 22 not specified). The sample included 577 pupils from the 8th grade, and 1227 and 1193 from the 9th and 10th grades, respectively (18 not specified). The response rates varied widely between schools, ranging from 6% to 69%; however, there were only a few schools where more than half of the pupils participated.

The data were analyzed by means of descriptive statistics using SPSS Version 29. After the analysis, we documented the findings in individual school reports that were sent to the respective schools. These reports addressed the most critical issues regarding the meal situation and provided recommendations for improvement.

In accordance with the transdisciplinary strategy, we obtained qualitative data by organizing different events that facilitated further reflection on the findings and provided opportunities to discuss solutions with schools and other relevant actors (Appendix A). The events were also used to collect more qualitative data on political and legal conditions, the relationships between the individual actors and their roles in the field of Berlin school catering.

The overall results of the survey were presented to the NCSC and the SenBJF in two online meetings. Subsequently, two networking events were held together with the NCSC in September and October 2022. The first event took place at the Center for Technology and Society. An invitation via email was sent to those schools in our survey which had a fully equipped kitchen. The second event was held online; all of the other schools were officially invited. In both events, we presented our findings to representatives from a total of 23 schools including principals, teachers, caterers, representatives from the municipal school authorities and parents. Based on their expertise on Berlin school catering as well as on nutrition, the NCSC led and moderated the workshops. Participants were asked to describe the lunch situation at their schools and to share their experiences. Due to the open format of this event, they could also address different subjects on their own such as lunch room and organization, the food quality or the pupils’ preferences and dietary patterns.

Additionally, three coaching sessions were held at individual schools during December 2022. We invited schools that participated in the previous networking events and expressed interest in improving the lunch situation at their school. The NCSC was mainly responsible for the organization and the moderation of the sessions. Between six and nine people attended each session including principals, teachers, pupils and parents. At the beginning of these sessions, we reported the survey results of the individual schools to the participants. These results pointed out key areas for improving the school lunch from the pupils’ point of view. With the expert input from the NCSC, different strategies were then developed and discussed with the respective schools for improving the situation.

Finally, the NCSC organized a workshop on vegetarian food in school catering. This workshop took place in a professional kitchen and a seminar room owned by the project “Kantine Zukunft”. Under the guidance of a chef, three Berlin caterers from schools in our survey were trained in cooking vegetarian dishes. The principal investigator was an active participant in this workshop and collected data via observations and informal talks with the attendants.

All of the aforementioned discussions and events were documented and transcribed in anonymized minutes and observation records by the researchers. Informed consent statements were obtained in advance. In addition, notes from several evaluation meetings with the NCSC in which we co-interpreted the findings were also considered in the qualitative analysis. Furthermore, documents, press releases and newspaper articles were reviewed in order to gather additional information on the catering situation in Berlin regarding recent meal programs, regulations and policies.

The qualitative analysis was guided by the key dimensions of the conceptual background (see Table 1). This was followed by an inductive coding approach through which we identified four tensions in the transformation of school catering. These findings were organized around the main aspects of the German catering system and, later on, also discussed in meetings with the NCSC.

The following section provides an overview on the formal roles and responsibilities in German school catering which will provide important context information and serve as a structure for the presentation of our results.

## 4. Roles and Responsibilities in School Catering: Overview of the Situation in Germany

The German system of school catering involves a significant number of actors, each of them with their own roles and responsibilities. Based on Jansen et al. (2020) [44], this section focuses on six key actor groups: (1) the legal authorities on federal and municipal level, (2) the caterers, (3) the schools (principals and teachers), (4) the pupils and their parents, and (5) intermediaries such as the NCSC.

Federal state authorities provide the legislation around school catering and are also responsible for the development of school meal programs and new regulations. The municipalities that operate under the broader legal framework determined by the federal school laws are responsible for the provision of school meals [45].

The task of providing school meals is usually outsourced to private caterers who participate in a tendering process to compete for a contract. The process is managed by the municipal school authorities, who also determine the tender specifications. These include, for instance, the technical and organizational requirements, the feedback procedures, the expected menu compositions and the prices of each meal. Compulsory dietary standards for serving healthy, sustainable and inclusive school meals do not exist in Germany [15]. Before making a decision, municipal school authorities enter into continuous exchange with the respective school counsels, ensuring that sufficient consideration is given to their specific needs. In accordance with EU public procurement regulations, caterers meeting the tender specifications with the “best price-quality ratio” [46] (p. 23) will be awarded a contract. Besides functional requirements, these specifications may also include social, health and environmental standards. The school authorities are also responsible for funding the construction and maintenance of (new) buildings for lunchrooms and kitchen facilities.

For the duration of the contract, the caterer is fully responsible for managing the meal service, including food preparation, menu planning, hygiene and serving food. In some schools, the caterer also operates the school kiosk or cafeteria where snacks and beverages can be purchased. The landscape of school caterers is quite heterogeneous in terms of the size of companies, cooking methods used, and the catering systems employed. If proper school kitchen facilities exist, some caterers offer fresh cooking, while others specialize in delivering food from an outlet kitchen [47].

Although they have no direct control over the organization of the meal service, schools can indirectly exert influence through school counsels or food committees. For instance, schools are often involved in the selection of caterers [44]. Moreover, schools have a certain degree of autonomy when it comes to deciding on the content and focus of teaching and learning concepts. This allows them to integrate topics related to school meals and healthy nutrition not only into school policies and practices but also into the curriculum.

Pupils are not obliged to participate in school lunch and are therefore free to choose whether, how and when they wish to eat. However, their decision making can be influenced by teachers and their parents [47]. Both can contribute to a food education that shapes the children’s eating behavior and general attitude towards food. In most cases, parents are responsible for paying for the meals unless they receive state assistance.

The NCSCs exist in all 16 German federal states and are usually funded by the national or federal state government [44]. As an intermediate actor, they operate under the umbrella of the National Quality Center for Nutrition in Daycare Centers and Schools. Their work encompasses the organization of educational programs that target the school community, but also counseling services aimed at improving the quality of school meals. In order to address systemic deficits, the NCSC network combines forces in formulating recommendations for political action on various issues such as the incorporation of dietary standards into the tendering processes [47].

Figure 1 summarizes the multifaceted system of German school catering including actors, responsibilities and fields of activities. The next sections present the main tensions from the analysis of our empirical data for Berlin secondary schools. The results will be presented along the relevant aspects in school catering that are marked by the orange boxes.

## 5. Results

### 5.1. School Law, Meal Programs and Tenders for School Catering: Lack of Public Funding and Catering Standards (Tension 1)

The document analysis and the exchange with the NCSC and the Berlin Senate showed that the city state of Berlin has made several attempts to improve the quality and dietary standards of school meals in the last two decades. This began back in 2003 with a quality improvement campaign that was launched in cooperation with the Berlin NCSC, the SenBJF and the AOK Berlin, a health insurance provider [48]. In 2013 this was followed up by the parliament’s adoption of a new school law on improving the quality of school lunches [49]. The aim was not only to introduce better standards in catering procurement but also to involve schools more closely in this process. Additional efforts were made in subsequent years, but most of them focused on improving the catering situation for elementary schools. For instance, in 2019, the Berlin parliament passed a law that guaranteed all elementary school pupils a free school lunch. Later on, the decision was taken to incorporate mandatory sustainability criteria into tender documents. This required caterers to cook with 50% organic food by 2021 [50].

The documents show that secondary schools do not receive any subsidies for catering from the city government, so budgeting issues make adequate quality improvements of school food environments more difficult. Unlike in the case of elementary schools, the city government does not plan to either provide additional funding or to implement compulsory standards that aim, for instance, at increasing the share of organic food. According to the NCSC, “any efforts to subsidize sustainable (and cost-effective) school meals have been put on hold by the Berlin Senate [in 2022]” (meeting, May 2022). The state-funded “berlinpass BuT”, provides the only possibility for children from vulnerable households to get a free meal, but parents can only apply for it if they receive social welfare payments.

Due to a lack of compulsory standards, municipal school authorities are responsible for including catering specifications in the tender documents. These specifications predetermine the provision of school food environments for the entire duration of the contract. According to the NCSC, the Berlin authorities normally use templates that are based on dietary recommendations from the German Nutrition Society. These templates were originally drafted and updated by the SenBJF in coordination with the NCSC, the Berlin districts and the federal parents’ committee [51].

The school law recommends the involvement of schools in the process of formulating catering specifications, ensuring that they match the individual needs and conditions of the school. Our empirical data showed, however, that this is not always the case. In fact, two of the schools that were supported with coaching had neither access to the contract nor knowledge about its key components and duration.

In the coaching sessions, some school principals also reported having considerable difficulty finding appropriate caterers due to the limited number of companies that offer these services in Berlin (networking event 1). It was argued that this lack of competition results from low profit margins and limited public subsidies, making public sector school catering a rather unattractive business model.

Despite the lack of public funding and catering standards, some political and intermediary actors are making efforts to improve the lunch situation within secondary education. For instance, the Berlin Senate Department for Justice and Consumer Protection (SenJV) was officially in charge of the development of the Berlin municipal food strategy in 2020. One goal of this strategy is to increase the amount of organic, regional, healthy and fair-trade food in all public catering facilities in the city [52]. The SenJV also finances the project “Kantine Zukunft” [53], which offers caterers training courses and seminars that deal with the different aspects of providing a more healthy and sustainable food service. With support from the SenBJF, the NCSC is another important actor that leads a number of programs such as the “IN FORM Projekt 2021–22” [54], which provides better food education to children and teachers. Another state actor is the “Serviceagentur Ganztag” (SAG), an agency that provides support and coaching to all-day schools, including coaching on catering. In a recent publication, they included school catering as an important pillar in the concept of all-day schooling [55].

### 5.2. Catering Service and Lunchrooms: Incompatible Demands and Preferences (Tensions 2)

According to the NCSC, providing healthy, inclusive and sustainable school meals is a difficult undertaking for caterers. Dishes must meet contract specifications while also matching pupils’ preferences.

The empirical findings of the quantitative survey indicate that caterers rarely manage to provide a food offer that achieves a high level of acceptance. As presented in Figure 2, the data show very low participation of pupils in school meals in Berlin secondary schools. Two thirds of the pupils (67%) reported that they never eat lunch at their school even though a warm meal is on offer. Only 21% eat a school meal at least 1–2 times per week (Appendix A).

On average, pupils tend to be dissatisfied with the menu, the presentation or the taste of the food (see Figure 3 or Appendix A). When pupils were asked in the survey about their preferences, many requested a greater variety of food, e.g., food that is more popular among their peers, such as fast food, but also food that encompasses the culinary traditions they grew up and identify with. For 22% of the pupils, the main reason for not attending school lunch was the taste of food.

In total, 12% of the children stated that they do not attend school lunch because it is too expensive for them or their parents (see Figure 3). Discussions with the schools indicated that the average meal price was between EUR 3.50 and EUR 4. While these prices were too high for some parents, they are too low for some caterers. One of them argued that they would barely suffice to keep their business afloat (Caterer 2, cooking workshop).

A total of 15% of the pupils stated that they would rather eat smaller snacks from the kiosk than having a full meal at lunch (see Figure 3). In all three coaching sessions, pupils and teachers also mentioned the popularity of snacks among pupils, but they also pointed out that most of these snacks were unhealthy. For instance, one pupil reported that “*many children regularly buy nugget rolls, soft drinks and iced coffee in summer from the kiosk*” (Pupil 1, coaching session 1). However, school meals must comply with the nutrition standards in the catering contract with the municipal authorities.

The data show that preferences and demands within the group of the pupils and other actors of the school community do not seem to be easily compatible. Caterers are therefore challenged by addressing those different demands in a way that their budget is not overstretched.

In the coaching sessions, healthy snacks such as vegetarian wraps or fruit salads were discussed as a possible replacement to be sold at the kiosk. However, the NCSC pointed out in later conversations that many caterers lack sufficient know-how or the necessary kitchen facilities to prepare healthy food alternatives in a proper manner (evaluation workshop, July 2022). There was also discussion around offering more vegetarian or vegan dishes that can be produced from plant-based ingredients, which are generally cheaper than meat. Reduced meat consumption is compatible with dietary standards, but at one networking event, the caterers expressed concern that even fewer pupils would attend school lunch if meat was excluded from the menu. To reduce the financial pressure on caterers without driving prices up, it was suggested that schools should improve their advertising of the lunch offer, while caterers should communicate improvements more effectively in order to increase the number of meals sold.

Another aspect of the school food environment is the lunchroom. Figure 4 and Appendix A show that most pupils do not enjoy eating in the lunchroom because of noise levels, lack of comfort, insufficient seating and cleanliness issues. In two of the coaching sessions, teachers also pointed out the limited capacities of their lunchrooms and the tight schedules, resulting in the introduction of restrictive attendance rules.

Children did not appreciate these rules and, as illustrated by the following survey response, complained about the fact that they are neither allowed to use their phone nor to remain seated after finishing a meal:
*“I wish we had more freedom and could eat normally, that means, use our cell phone sometimes (of course, without sound). I hope that we can stay in the cafeteria with our friends even though not everyone is eating.”*(Pupil 1, survey)

In a coaching session, one school reported employing a less restrictive strategy in order to reduce the amount of time children spent in the lunchroom and maximize the usage of space by allowing pupils to serve themselves from bowls on their tables (coaching session 3)

There was a wide consensus among the schools and the NCSC that cafeterias should be turned into more enjoyable places for social encounters for children, where they can relax and interact with each other while eating. However, the physical structures and the rooms’ interiors cannot be changed easily. Changes require lengthy approval procedures from city authorities, who are in charge of the (re)construction, financing and safety of current and new buildings. As a consequence, bureaucratic processes can hold up changes to lunchroom environments that are already underway. For example, a teacher in one of the coaching sessions complained that the school was promised an extension of the cafeteria building, but “*for three years nothing has happened*” (Teacher 1, coaching session 2).

According to the NCSC, schools can only implement minor and affordable changes to improve the lunchroom atmosphere, such as decorating the room or rearranging the chairs and tables. Engaging pupils in this work was seen as a possible way for struggling schools to maintain a certain level of cleanliness. As a caterer explained at one networking event, vandalism is a common problem—pupils damage the furniture or litter the cafeteria.

### 5.3. Food Strategies, Teaching and Learning Concepts: Lack of Resources and Opportunities for Complementary Education and Participatio (Tension 3)

Schools can integrate food strategies into the all-day school concepts, using them as a starting point for a renewed strategical orientation that focuses on a common vision with defined goals and distributed responsibilities [47]. However, the schools reported that problems such as the poor quality of school lunches could not be properly addressed due to limited time and personnel resources. In one of the workshops, we also discussed the role that teachers, in their function as all-day coordinators, can play in providing better school food environments. However, it became clear that the all-day coordinators who were assigned to these tasks were neither being paid for any extra working hours nor being partially released from their teaching duties. Within the requirements of the federal state’s curriculum framework, schools and teachers are able to integrate the topic of school food into formal teaching and learning concepts and to structure school classes accordingly. The NCSC emphasized how food-related topics can come under focus within various courses, such as biology, geography and finance. They can also be taught more thoroughly in “Economics-Work-Technology” classes, which are already part of the curriculum in Berlin secondary schools. Although pupils are introduced to healthy food consumption and food processing, there is limited time to discuss these topics in more detail due to the broad scope of this subject.

The NCSC emphasized the potential of schools to contribute to the formal and—more importantly—informal food education of children. Informal learning was discussed as an outcome of various interactive activities in school. This may include involving pupils in the menu planning, cooking in the school kitchens, or table service. At some of the schools, for example, pupils’ enterprises are responsible for managing the cafeteria, which is a very far-reaching approach to informal learning.

A number of participants in the networking events acknowledged that informal learning is particularly promising because pupils tend to learn more easily through their own experience. Informal learning can occur as an outcome of the daily food routines that are part of the school culture. For instance, one of the schools emphasized how teachers and the principal became role models by eating in the cafeteria on a regular basis (networking event 2). According to some of the teachers, because pupils take notice of the school staff’s physical presence in the lunchroom, they may feel encouraged to join them. The routine of teachers going to lunch together with their class was discussed in the networking events and the coaching sessions as a chance to increase school meal acceptance. However, these routines are not well-established in most of the schools we worked with. In fact, 78% of the pupils in our survey disagree or rather disagree with the statement that they eat lunch together with the class (see Appendix A). Some teachers were hesitant to take up this initiative, as they valued their free time and neither the food nor the eating experience was particularly appealing to them from a personal point of view.

The findings show that schools can foster participation by setting up food committees. Teachers, caterers, pupils and parents can be members of these committees. A key task assigned to these committees is food quality control [56]. At the committee meetings, mutual expectations and potential improvements regarding the school food environments can be discussed in direct exchange with caterers. If contract terms are violated by caterers, municipal school authorities will be informed.

Despite the legal obligation imposed by the federal school law [57], food committees were not implemented in all of the schools that participated in our research. Their work was not considered to be feasible for every school concept, and its practicality was questioned. For instance, one teacher anticipated low motivation among her colleagues, who would not wish to participate due to their individual workload and the fact that the school does not have any compulsory afternoon classes (coaching session 1). Another challenge mentioned by one teacher is the potential under- or overrepresentation of certain groups of pupils on these committees.

An important reason for pupils not going to lunch is a lack of opportunities to participate. In almost all of the schools in the study, pupils reported that there are few opportunities to get involved in any feedback or decision making processes (see Appendix A). For instance, the majority of pupils cannot give regular feedback to the caterers (93%, n = 2326, see Appendix A). Findings from the survey also show that most pupils (59%) do not or somewhat do not trust the caterer at their school (see Appendix A). Table 2 shows that some pupils were interested or rather interested in opportunities to vote for their favorite meals (45%) or to evaluate dishes and give feedback to the caterer (31%).

The pupils’ interest in being involved in the decision making process is also reflected in the survey’s open answers. For instance, one pupil mentioned how important it is “*to have a say in the school meals and that food preferences are taken into account*”. (Pupil 2, survey).

In the coaching sessions, the NCSC emphasized the positive effects of providing opportunities for pupil feedback, as they anticipated this could lead to greater satisfaction and thus increase the number of pupils who eat regularly in school. Yet, according to the NCSC, larger caterers who deliver similar standardized menus to a considerable number of schools might not be able to respond to individual feedback.

In other discussions, some school representatives were skeptical of pupil participation. They believed that opportunities for feedback might promote unrealistic expectations among pupils, especially if they do not understand the conditions under which caterers need to operate. As a solution, the NCSC recommended allowing pupils to choose dishes for the week from a preselected menu provided by the caterer.

### 5.4. Eating Outside of School and Parental Food Education: Peer and Parental Influence (Tension 4)

According to Figure 3, a certain percentage of the pupils prefer to eat at home (23%) or to eat packed lunches brought from home (22%) instead of participating in the school lunch. Pupils reported consuming smaller snacks because they routinely eat a large dinner with their family after school. Teachers and the NCSC showed considerable understanding for the tradition of having dinner with the family. Then again, it was pointed out by the NCSC in one of the coaching sessions that if children skip lunch or exclusively eat unhealthy snacks in school, their nutritional intake throughout the day may be insufficient.

Figure 3 also indicates that another reason for children not taking up school meals is that their friends choose not to do so (13%). Although children are not officially allowed to leave school premises during lunch, 10% of them regularly eat outside of the school grounds. A school principal in the first coaching session mentioned that this is problematic in Berlin because of the wide selection of fast food offers and grocery stores in the city, which are very popular among adolescents. Parents also play an important role in school catering. Given that parents sign the contract with caterers and pay for the meals, it may be assumed that both parents and children need to be convinced that the food is worth the price in terms of taste and quality. For example, one parent expressed his disappointment with the caterer as follows: “*The quality of the food is poor and it is not tasty. […] Although the caterer prepares food in a school kitchen, financial aspects are his main priority*” (Parent 1, coaching session 3).

Furthermore, not all parents seem to be sufficiently involved in their children’s food choices. For instance, one of the teachers in the coaching sessions raised the issue that some parents simply give money to their children without showing much concern for what they actually eat in school.

In the coaching sessions, a proposal was made to increase parents’ involvement by offering more information that is easy to understand and focuses on the advantages of participating in school lunch. A particular focus was placed on promoting inclusive food access. To this end, schools were asked to encourage eligible parents to apply for the “berlinpass-BuT”.

The survey findings indicate that of the proportion of children who have a valid pass (37%), less than half (40%) of them actually use it for the warm school lunch (see Appendix A). Based on these findings, the NCSC emphasized that the information provided to parents should also include reference to the application process for free meals, but more importantly, encourage them to actually make use of this offer. Pupils in receipt of free meals can be seen as a reliable source of income for caterers, since their daily participation in school meals is significantly higher than the average.

## 6. Discussion

Our findings highlight the challenges of transforming school catering. These arise from tensions between actors at the federal state, municipal, school and private levels. The first part of this section reflects on the added value of referring to the concepts of food environments and social cohesion in analyzing the food situation in German schools. This is followed by a discussion of recommendations for navigating the transformation of school catering along the four major areas of conflict identified in the study.

### 6.1. Reflections on Applying the Concepts of Food Environments and Social Cohesion

The concept of food environments accounts for a number of different physical, economic, regulative, and social dimensions, and thus contributes to a comprehensive understanding and assessment of school catering and its multifaceted components [21,22]. From this perspective, unhealthy eating in school is understood as eating habits that are shaped and reproduced by environmental conditions and group dynamics.

With the addition of the social cohesion concept, we explored the social aspects in food environments in detail. Firstly, through analytical differentiation, it was possible to focus on the socio-physical interrelations within food environments. For instance, we were able to show how lunchrooms can become a barrier to developing supportive social norms of eating together or how public food programs can contribute to inclusive offers by lowering meal prices. Conversely, the popularity of junk food can be explained by convenient access and prevailing consumption norms in peer groups. Secondly, the concept of social cohesion points towards the challenging processes of shaping school food environments due to the limited scope of action for public and non-public actors and the fragmented division of tasks. Therefore, creating school communities that enable coordinated action and inclusive participation might be just as important in encouraging children to eat healthier as increasing the accessibility of healthy food [24]. However, this approach often goes along with tensions between relevant actor groups and their interests and expectations. For instance, the goal of strengthening a school community by including everyone may overlook pupils’ social needs in terms of developing their own preferences and engaging with their peers. Thirdly, by connecting the two concepts of food environments and social cohesion, we were able to conduct a multi-perspectival analysis that suggested the importance of developing integrative strategies for a comprehensive transformation of school catering. Instead of improving school meal consumption by focusing on changing elements of food environments [27], integrative strategies encompass the simultaneous introduction of interventions by multiple actors operating on different levels within school catering. These are interventions that take into account a large number of (sometimes conflicting) factors, rather than focusing on single-target modifications [23].

### 6.2. Recommendations for the Transformation of School Environments

The following recommendations address the tensions identified in the study that are a consequence of: (1) the lack of public funding and catering standards, (2) incompatible demands and preferences, (3) the lack of resources and opportunities for complementary education and participation, and (4) peer and parental influence.

#### 6.2.1. Public Funding and Catering Standards

The results underline the importance of defining compulsory tender criteria on the basis of municipal procurement policies. Examples from Italian municipalities point out how this type of criteria can improve food quality [4,34]. In light of rising food costs, however, caterers might struggle to meet these standards and decide to exit the market, which lowers the chance of schools finding suitable providers. Meal programs that aim to promote healthy and sustainable eating in schools may, therefore, provide additional funding for proper adjustments. Municipalities could also aim to implement higher standards gradually, which would allow caterers time to adapt. In Berlin, tenders for elementary schools also aim for a steady increase in organic food. The required percentage increased from 15% to 50% over several years [50]. Moreover, expanding free meal programs helps to reach vulnerable children from low-income families who do not receive social security payments, and provides them with access to valuable nutrition [11]. Attention needs to be paid to potential trade-offs of free meals, such as increased food waste and costs per meal due to revenue cuts, as well as the potential stigmatization of children living in hardship [11,12].

#### 6.2.2. Incompatible Demands and Preferences

Our findings show that two thirds of the pupils never eat lunch in school. Other studies on German secondary schools show similar results or slightly higher participation rates—between 33% [17], 37% [15] and 58% [16]—but these results include pupils from lower secondary levels.

Our analysis also confirms that meal participation among pupils is closely connected with the menu, the organization of the meal service and the quality of the food [15,17]. But aside from these general aspects, our results point to the importance of dietary preferences based on lifestyle and religious motivations. Pupils might not accept sustainability interventions if they are not compatible with their individual diets. For instance, if unhealthy snacks are no longer offered, pupils may decide to buy them from competing food outlets outside school instead [23].

Caterers need to increase their proficiency in specific diets and their respective preparation methods (e.g., halal, vegetarian, vegan and low-calorie diets) in order to add more variety to the menu. The foundation of food committees holds great promise to find necessary compromises and resolve potential conflicts between pupils’ preferences, catering requirements and nutrition standards [34]. This might be particularly relevant in city schools such as in Berlin, which are characterized by high socio-cultural heterogeneity.

#### 6.2.3. Lack of Resources and Opportunities for Complementary Education and Participation

Schools can take a leading role in linking up municipal school authorities and caterers on the one hand, and pupils and parents on the other. Networking actors such as the NCSC can assist as mediators in building these partnerships, contributing their expertise in school meal regulation and their thorough understanding of different actors’ perspectives and priorities, which may be conflicting at times. The formulation of a school food strategy can establish a common ground for these processes. The NCSC can also facilitate partnerships beyond the school context with actors such as cooperatives, community initiatives or NGOs. Larger networks can support schools in nutritional education (e.g., by offering excursions to local farms), finding new caterers, or gaining access to organic food supplies from local sources.

Results show the importance of an environment that values and acknowledges feedback from pupils, even if following up on their suggestions is not always possible. The findings indicate that teachers are key actors when it comes to encouraging pupils to get involved. Under the premise of more favorable working conditions, teachers could be trained to use their influence as role models more strategically and with greater awareness [29].

Furthermore, food education may be included in the curriculum, but it also needs to become an integral part of the school culture and practices that go beyond classroom teaching. This could mean, for instance, offering cookery classes or gardening projects allowing pupils to gain experience, develop skills and thereby nurture their interest in this topic [40]. All-day schools may have the necessary time resources in the afternoon hours to offer additional courses and extracurricular activities that foster informal learning.

#### 6.2.4. Peer and Parental Influence

The results are consistent with previous studies demonstrating that adolescents prefer to eat in school with friends and perceive this as a leisure-time activity [28]. While it is not feasible to make structural changes to school buildings on a short-term basis, lunchrooms can be turned into more welcoming spaces that facilitate peer interactions through minor modifications such as wall decorations, or by rearranging the seating.

Finally, the role of parents should not be overlooked, as they can have a strong influence on whether their children consume meals in school [36]. Some parents, for example, might not be convinced by the quality of the food, while others may want to retain control over how their children are fed [39]. Reinforcing teacher–parent relationships that foster mutual communication helps to address these conflicts. This could lead to a change in the parents’ perception of the quality of school meals. Better communication can also make it easier to inform parents who struggle with meal costs that free lunch options are available.

### 6.3. Limitations of the Study

A certain number of limitations need to be considered when interpreting the findings. Even though Berlin as a city state is a special case, the organizational structures for school catering are comparable throughout Germany. Therefore, we would argue that, to a certain extent, the findings of this case study also apply to school catering in other German federal states. However, due to the considerable differences in the organization of school catering in different countries, it is not possible to generalize beyond the German context. Nonetheless, the literature indicates that many countries face a number of similar challenges, including insufficient funding, unhealthy and non-sustainable food offers and low take-up rates. Moreover, our sampling is not fully representative of secondary schools in Berlin, since instead of drawing a sample in advance, all of the existing schools were invited to participate in the survey. Nevertheless, the sample contains a wide selection of schools of various sizes that are located in urban districts with different socio-economic characteristics. Our findings might also underestimate the actual participation rate of pupils due to the effects of the COVID-19 pandemic. Regarding the transformation process, additional qualitative case studies might contribute to an understanding of how to rebuild school food environments within school communities by exploring in greater depth the roles of the actors involved.

## 7. Conclusions

School catering has the potential to shape the dietary habits of future generations positively and contribute to public health, sustainable eating and inclusive food access. This means it has transformational potential, but addressing multifaceted school food environments and key actors is a challenging endeavor. Applying the two concepts of food environments and social cohesion offers important insights into the complexity of transforming school food environments.

We suggest integrative strategies that combine programs and policies in a top-down direction, and the development of interventions within school communities. This implies additional public funding and requires political action from national and municipal actors. There should be greater social recognition of school catering in terms of its wider benefits for children and schools as well as its contribution to public welfare. If school caterers received financial compensation for adopting higher health, social and environmental standards, they would be less dependent on the dominant principle of cost efficiency.

Top-down strategies have to be complemented by bottom-up activities on the level of the school communities. This means strategies that are built on strong partnerships, higher levels of involvement, and integration into everyday school culture and routines. Intermediate actors and integrative strategies that establish long-standing cooperation between school community members play an important role in this process. Particular attention needs to be paid to the perspectives of pupils, as it is they who decide what, when and how they eat in school. There is also much to learn from best-practice examples in schools that have already taken the initiative and are on their way towards a transformation of school catering. It is also worth taking into consideration effective practices, supportive measures and actors’ experiences of school meal policies in other European countries, as these offer points of comparison for transforming the school meal situation in the German context.

## Figures and Tables

**Figure 1 ijerph-21-00370-f001:**
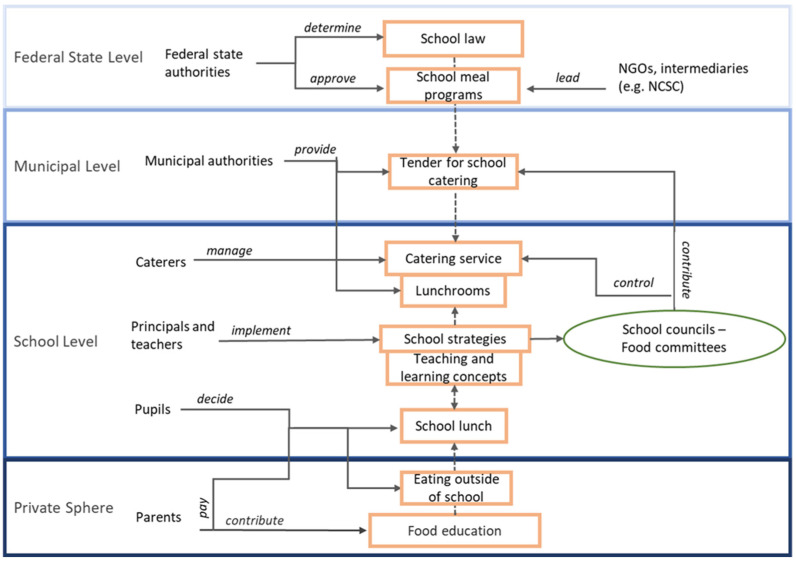
The German system of school catering.

**Figure 2 ijerph-21-00370-f002:**
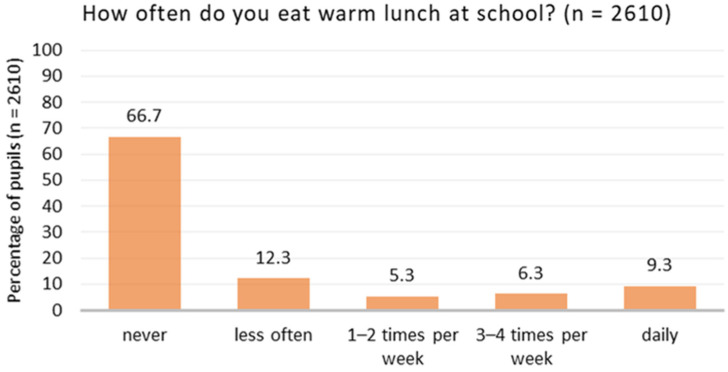
Frequency of participation in warm school lunch (*n* = 2610).

**Figure 3 ijerph-21-00370-f003:**
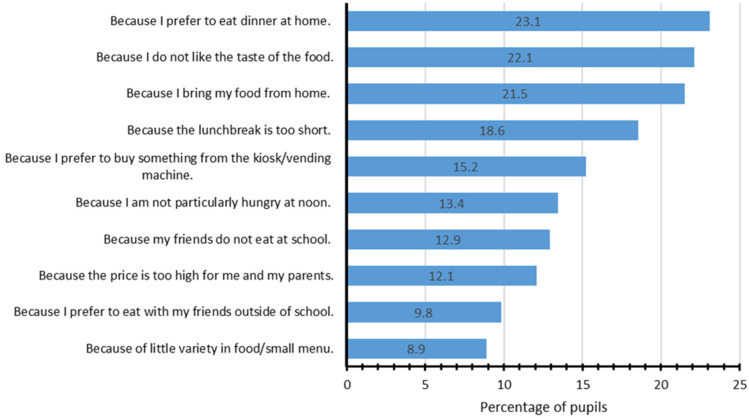
Reasons for never or rarely eating a warm lunch at school (multiple answers).

**Figure 4 ijerph-21-00370-f004:**
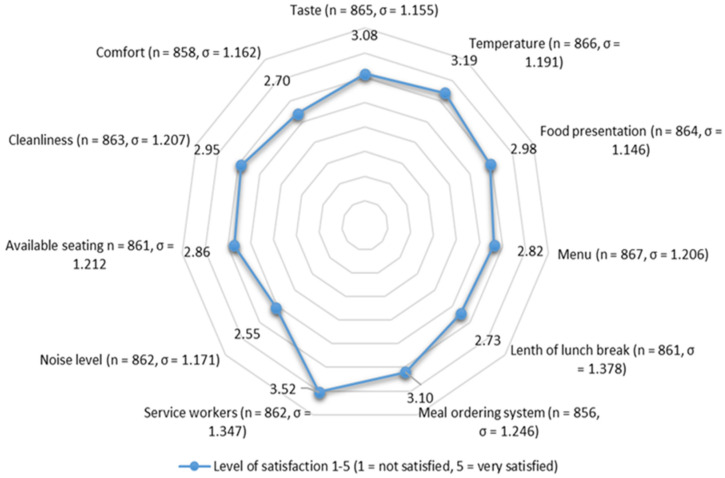
Satisfaction with the food and the lunchroom.

**Table 1 ijerph-21-00370-t001:** Conceptual background for analyzing school food environments and school communities.

Dimensions	Sub-Dimensions
Physical	food options, kitchen infrastructures, lunchrooms, competing food services
Economic	meal prices, catering and food preparation costs
Regulative	legal framework, meal regulations, dietary standards, food service organization
Socio-cultural	food habits and tastes, role of teachers, social norms of peers
Social cohesion	quality of interactions between different actors within school catering
	opportunities for participation (e.g., attending school meals, participating in working groups, committees)
	inequalities (income, socio-cultural background) and inclusiveness

**Table 2 ijerph-21-00370-t002:** Pupils’ interests in participation opportunities.

Would You Be Interested in the Following Activities and Participation Opportunities?						
Voting for favorite meals	1 = Not interested	2	3	4	5 = Very interested	Total
n	459	302	632	473	646	2512
%	18.3	12.0	25.2	18.8	25.7	100
Giving feedback to the caterer	1 = Not interested	2	3	4	5 = Very interested	Total
n	537	359	829	427	358	2510
%	21.4	14.3	33	17	14.3	100

## Data Availability

The data presented in this study are available on request.

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
