# Peer review of "Healthy, Inclusive and Sustainable Catering in Secondary Schools—An Analysis of a Transformation Process with Multiple Tensions"

_ijerph, 2024, doi:10.3390/ijerph21030370_

Round 1

Reviewer 1 Report

Comments and Suggestions for Authors

It appears that the authors are describing the the development and completion of a pupil survey followed by a stakeholder consultation process to improve the school food environment. If that is the case, it is very hard to draw out research objectives, study design and methods, and the results of the work (including the survey design, results of the survey and qualitative methods). As such, the article needs to be much more transparent with methods and results adequately reported. The results also read more like a discussion piece or literature review and should focus more on presenting the findings (supported by empirical data).

Abstract

Methods describe that data were collected via survey and workshops, but no data analysis methods provided.

Results appear to only describe qualitative findings – what are the findings of the survey?

Methods

Line 149 – as the study follows a transdisciplinary research strategy, more detail is required to explain how this strategy worked in practice. For example, what approach was used to develop the pupil survey? Were previous examples drawn upon? How were questions agreed among stakeholders? Was the survey piloted? Is the survey available as an additional file that can be used by others?

Line 162 – can you provide characteristics of the schools that responded?

Line 174 – can participant characteristics be provided?

Line 181 – which variables were tested for correlation – were these decided upon in advance or after the data had been reviewed

Line 187 – 199 as these meetings/ events were the source of all qualitative data, more detail is needed. For example, who ran the event/ meeting? What were the skills and experience of the people running the meetings / events? How were results of the survey presented to stakeholders? how were stakeholders invited to attend? What format did the events/meetings take? What questions were stakeholders asked about the results? What was the relationship between researchers and stakeholders? Can session plans be provided as an additional file? - refer to COREQ checklist

Line 196 – more detail is needed to understand how improvements were planned and implemented? Was this part of the research or an aside (if so, not relevant to the methods here)

Lime 204 – more detail is needed to describe the thematic analysis. Who undertook the analysis? What approach was used (deductive / inductive). How were key themes agreed? Were findings presented back to stakeholders? – refer to COREQ guidelines

Line 207 – section 4 – I assume this section has been included to detail the constructs in which the results are presented – please explain how this fits with the thematic analysis approach that was undertaken

Results

The way the results are currently presented does not comply with the format of an original research article. It reads more as a literature review or discussion paper with some empirical findings included to support the narrative. For example, no empirical findings appear to feature until line 313. I would expect results of the survey to be presented first (or provided as an additional file) and then the qualitative discussion of the survey results to be presented next, according to key themes and with supporting quotes or fieldnotes.

What was the outcome of the thematic analysis - which key themes and sub themes were identified?

There appear to be many findings declared that are not suitably backed up by supporting data. For example, line 420 “The routine of teachers going to lunch together with their class also contributes to school meal acceptance, according to our findings. For instance, 35% of 421 the pupils who reported eating together with their class participate in school lunch on a daily basis, compared to 25% who reported never eating with their class”.  How do these findings prove that 35% of pupils eating lunch on a daily basis is linked to whether or not teachers are present? Did you test this hypothesis? Do qualitative data suggest that this is the case?

There is a lack of transparency around where results have been derived from which reduces trust in  the work. For example: Line 456  - The results showed that opportunities to evaluate dishes or vote for their favourite meal were highly appreciated by pupils – which results, where is the data to support this?

Reviewer 2 Report

Comments and Suggestions for Authors

Dear Authors,

The authors conducted a study on the school diet of secondary school pupils in Berlin. The research undertaken is part of the discussion on proper nutrition for young people, as well as the elimination of malnutrition among children from less affluent families. The topic undertaken fits into the scope of the IJERPH. The positive aspects of the manuscript are:
- a large and diverse population, with roughly equal numbers of opposite sexes,
- results based not only on questionnaires (which are always declarative), but also on interviews and conversations,
- survey questions consulted with state agencies working in the area of mass catering supervision.
However, the authors should address the following issues:
- I have doubts that the description of the structure of the manuscript is necessary (last sentence of Introduction); usually, scientific articles do not include such descriptions,
- the purpose of the survey is not clearly stated (it should be included at the end of Introduction),
- what do the authors mean by “inclusive catering”?; this is accentuated in lines 140-142, but should be explained in a little more detail,
- surveys carried out during pandemic C19 may have reduced the number of people potentially surveyed, as well as their food choices,
- no survey form attached as an annex or supplementary material.

Detailed editing notes:
108 [34–35] The > [34–35]. The
126-127 (Table 1 should be inserted immediately after the paragraph in which it is referenced. Headings in this table should start with a capital letter.)
177 grades respectively > grades, respectively
209 Jansen et al. (2020) > Jansen et al. (2020) [47]
212 Networking Center for School Catering (NCSCs) > NCSCs (The name of this institution together with its abbreviation has already been given in line 147.)
264 (Move title of the section to the next line.)
324-330 (Text should be the correct width and justified.)
357 banned > excluded
449 (Move title of the subsection to the next line.)
599 6.2.3 > 6.2.4
599 (Move title of the subsection to the next line.)
The syntax of some sentences seems incomprehensible. Therefore, I believe that the manuscript should be re-read by the authors with attention to this aspect and some sentences should be rewritten. 

Comments on the Quality of English Language

The syntax of some sentences seems incomprehensible. Therefore, I believe that the manuscript should be re-read by the authors with attention to this aspect and some sentences should be rewritten.

Round 2

Reviewer 1 Report

Comments and Suggestions for Authors

While this paper has improved from the previous version, the qualitative elements are still not adequately described or backed up by data. The main findings of this work are described as identifying key tensions / themes, but the data are not presented according to these themes, so there is no transparancy around where these themes have bee derived from. Currently, data are summarised according to Figure 1, with no evidence of thematic analysis taking place. The authors need to decide whether they want to present data in its current state (in which they should leave out mention of thematic analysis / key themes and tensions) or present data according to these  key themes / tensions with supporting data:

Specific comments below:

Line 227 the data were coded in an inductive and deductive was, using codes derived from our conceptual framework – I think this is the first time mentioned – which conceptual framework is this – Table 1? But your results are presented according to Figure 1?

Line 227 – 23 It still isn’t clear how the thematic analysis was undertaken. You say you have identified 4 key themes, but data are not presented / described according to these themes. Instead, you have presented data according to your framework presented in figure 1. It seems that you have just summarised your data according to each construct within the framework? The COREQ checklist says that data were deductively coded according to the two concepts of food environment and social cohesion, but data are not presented within these overarching themes either? If your main outcomes are related to your identification of major areas of conflict, why are the findings not presented as such to support your analysis?

Results – section 5.1.1 – As this is a results section it is still slightly confusing to present background information is provided here. Why not premise with, the literature tells us…. or the documentary analysis revealed that….

Line 363 – comparison to other studies would normally be presented in discussion
